# The Association of White Blood Cells and Air Pollutants—A Population-Based Study

**DOI:** 10.3390/ijerph18052370

**Published:** 2021-03-01

**Authors:** Shih-Chiang Hung, Hsiao-Yuan Cheng, Chen-Cheng Yang, Chia-I Lin, Chi-Kung Ho, Wen-Huei Lee, Fu-Jen Cheng, Chao-Jui Li, Hung-Yi Chuang

**Affiliations:** 1Department of Public Health, College of Health Sciences, Kaohsiung Medical University, Kaohsiung City 807, Taiwan; hsc0901@cgmh.org.tw (S.-C.H.); u101862003@kmu.edu.tw (H.-Y.C.); d770163@kmu.edu.tw (C.-K.H.); 2Department of Emergency Medicine, Kaohsiung Chang Gung Memorial Hospital and Chang Gung University College of Medicine, Kaohsiung City 807, Taiwan; malee4950@cgmh.org.tw (W.-H.L.); s12273@cgmh.org.tw (F.-J.C.); chaojui@cgmh.org.tw (C.-J.L.); 3Departments of Occupational Medicine and Family Medicine, Kaohsiung Municipal Siaogang Hospital and Kaohsiung Medical University, Kaohsiung City 807, Taiwan; u106800001@kmu.edu.tw; 4Department of Occupational Medicine, Kaohsiung Municipal Ta-Tung Hospital and Kaohsiung Medical University, Kaohsiung City 807, Taiwan; chiai@kmu.edu.tw; 5Department of Occupational and Environmental Medicine, Kaohsiung Medical University Hospital, Kaohsiung City 807, Taiwan; 6Department of Public Health and Environmental Medicine, College of Medicine, and Research Center for Environmental Medicine, Kaohsiung Medical University, Kaohsiung City 807, Taiwan

**Keywords:** air pollutants, white blood cell count

## Abstract

The links of air pollutants to health hazards have been revealed in literature and inflammation responses might play key roles in the processes of diseases. WBC count is one of the indexes of inflammation, however the l iterature reveals inconsistent opinions on the relationship between WBC counts and exposure to air pollutants. The goal of this population-based observational study was to examine the associations between multiple air pollutants and WBC counts. This study recruited community subjects from Kaohsiung city. WBC count, demographic and health hazard habit data were collected. Meanwhile, air pollutants data (SO_2_, NO_2_, CO, PM_10_, and O_3_) were also obtained. Both datasets were merged for statistical analysis. Single- and multiple-pollutants models were adopted for the analysis. A total of 10,140 adults (43.2% males; age range, 33~86 years old) were recruited. Effects of short-term ambient concentrations (within one week) of CO could increase counts of WBC, neutrophils, monocytes, and lymphocytes. However, SO_2_ could decrease counts of WBC, neutrophils, and monocytes. Gender, BMI, and smoking could also contribute to WBC count increases, though their effects are minor when compared to CO. Air pollutants, particularly SO_2_, NO_2_ and CO, may thus be related to alterations of WBC counts, and this would imply air pollution has an impact on human systematic inflammation.

## 1. Introduction

Owing to global urbanization and industrialization, the air pollution situation has been going from bad to worse in recent decades. The hazardous impacts of air pollution on human health have been perceived in nations across the globe. The World Health Organization (WHO) estimates that about one of every eight deaths in the world is air pollution-associated [1]. The Global Burden of Diseases, Injuries, and Risk Factors Study in 2016 reported about 4.1 million premature deaths and 105.7 million disability-adjusted life-years (DALY) worldwide could be attributed to exposure to ambient PM_2.5_ [2]. Also, the International Agency for Research on Cancer (IARC) has classified outdoor air pollution as a Group 1 carcinogen [3]. Studies have revealed the associations between air pollution and mortality and morbidity [4,5,6,7,8,9], and have established many links to different diseases, such as chronic obstructive pulmonary disease, cardiovascular disease, stroke, diabetes mellitus and different malignancies, etc. [10,11,12,13,14,15,16]. The possible responsible pathogenic mechanisms are still uncertain. Besides oxidative stress and genetic and epigenetic pathways, inflammation is one of the commonly accepted postulated mechanisms [17,18,19,20,21,22,23,24,25].

Air pollutants come from a variety of sources. These substances are suspended in the atmosphere, and can exist in the form of solid particles, liquid droplets, and gases. Among pollutants, particulate matter (PM) has been linked to the risk of respiratory and cardiovascular diseases [10,11,12,26]. PM affects human health differently depending on its size. When humans inhale PM, the large particles (>10 μm in diameter) would be trapped in the mucus lining of the upper airway (nose, trachea). If the diameter of inhaled particles is smaller than 10 μm, then they can reach the lower respiratory tract. Fine particles (PM_2.5_, with a diameter smaller than 2.5 μm) and ultrafine particles (UFP, with a diameter smaller than 0.1 μm) can reach the alveoli deep inside the lungs, and even penetrate through the epithelia into the bloodstream [23,27,28]. The inhaled pollutants cause epithelium injuries and activate the immune system to launch a cascading inflammation response [22,23,29,30].

White blood cell (WBC) counts in peripheral circulating blood is one of the indexes of the systematic inflammatory response of the human body. The count value is easily assessable and does not require an invasive procedure or expensive apparatus. Therefore, numerous studies use it as their investigation object. However, the evidence of air pollutants-associated leukocytosis in the literature is inconsistent. Tan et al., investigated 30 healthy adults who had been exposed to forest fires, discovering that air pollution was associated with the percentage of polymorphonuclear leukocytes, but was not associated with total WBC count changes [31]. Steenhof et al. investigated 31 healthy volunteers’ pollutant exposure at five different locations (two traffic sites, an underground train station, a farm and an urban background), and found that PM exposure was positively associated with the total WBC, neutrophil and monocyte counts; NO_2_ exposure was negatively associated with lymphocytes and eosinophils [32]. Stiegel et al. investigated 15 healthy volunteers, and found decreases in monocytes and lymphocytes and increases in neutrophils after exposure to diesel exhaust and ozone, but the total WBC counts seemed not to be unchanged [33]. Schwartz analyzed datasets (health data from The Third National Health and Nutrition Examination Survey, and air pollution data from the Aerometric Information Retrieval System), and found that PM_10_ exposure was associated with an increase of fibrinogen, and counts of total WBC and platelet [34]. Rückerl et al. conducted a prospective study involving 57 patients with coronary heart disease, and discovered that ultrafine particles (PM_0.1_) and fine particles (PM_2.5_) were associated with decreased total WBC counts, and PM_10_ seemed not to change the total WBC counts [35]. Poursafa et al., conducted a cross-sectional study among 134 school students, and found PM_10_ exposure was positively associated with total WBC counts [36]. Pope et al. investigated 24 healthy, non-smoking, young adults during three study periods, and found that the exposure to PM_2.5_ was associated with increases in the counts of monocytes, natural killer cells, helper T cells, and killer T cells [23]. On the other hand, Jacobs et al. investigated 38 healthy cyclists with exposure to 20 min traffic near a major road, and discovered that the PM exposure increased the percentage of neutrophils, but did not change the total WBC counts [37]. Zuurbier et al., investigated 34 healthy adult volunteers with exposure to PM during a 2 h commute, and found that air pollution exposure was not associated with changed of total WBC counts, but had a negative association with the counts and percentage of neutrophils [38]. Rich et al. using a quasi-experiment during the Beijing Olympics (a period of substantially decreased concentrations of PM and gaseous pollutants), found that the total WBC counts were not significant improved in that period [39]. In the literature, most related researches have small sample size, and this might explain the mixed results across studies.

We hypothesize that the peripheral circulating WBC counts, as one of the indexes of systematic inflammatory response, would still be associated with ambient air pollutants in a larger study. To exam the hypothesis, we conducted this population-based study. The aim of this research is to analyze the associations between multiple air pollutants on circulating white blood cell counts.

## 2. Materials and Methods

Kaohsiung City, with mountains, rivers, sea, long coastline, and harbors, is located in southern Taiwan. The metropolitan area has a population near 3 million, and the urban area is nearby the Kaohsiung Harbor. Its air quality is one of the worst areas in Taiwan because of the many industries located there. The governmental authority has set up air quality monitoring stations since 1993. Cianjhen (population about 201,000 in the 2003–2004 study period), Nanzih (population of about 161,000 in the study period), Hsiaokang (population of about 153,000 in the study period), and Cijin (population about 29,000 in the study period) are the four major monitoring stations which operate 24 h a day and cover more than 556,000 people. Hourly measurements of environmental pollutants collected at these monitoring stations include particulate matter of 10 micrometers or less (PM_10_), sulfur dioxide (SO_2_), carbon monoxide (CO), nitrogen dioxide (NO_2_), and ozone (O_3_). Air quality monitoring station, set up by the Taiwan Environmental Protection Administration and Kaohsiung City Government, are in Cianjhen, Nanzih, Hsiaokang, and Cijin districts. Therefore, this population-based study recruited subjects from these four districts from 2003 to 2004.

The government has promulgated the Air Pollution Control Act since 1975 and has set standards of air quality since 1992. The standards of air quality are reviewed and revised every 4 years. The air quality standards at that time were: yearly average emission for SO_2_, NO_2_, O_3_, CO and PM_10_ should be less than 0.02, 0.03, 9, 0.06 ppm and 50 μg/m^3^, respectively. The detail information can be referred to: https://oaout.epa.gov.tw/law/EngNewsList.aspx (assessed on 28 February 2021)

Kaohsiung Municipal Siaogang Hospital administered the clinical and laboratory examinations for study subjects with provided by the Kaohsiung City Government Department of Health. We selected study districts where air quality monitoring stations were located, and then randomly sent out invitations for volunteer participation according to household registration of the selected districts. Adult residents of these four districts were stratified-sampled by proportion of population, and then invited to participate in this health survey program by letter and telephone. Participants were volunteers who responded to the invitations. The exclusion criteria included pregnancy, current malignancy diseases, history of auto-immune diseases, and other possible infectious conditions (such as common cold) or fever on the exam day, and age < 30 years-old. The procedure was showed as the Figure 1. Demographic data including habitual behaviors were collected during the visit by questionnaire. In addition, the lifestyle (such as tobacco smoking, alcohol and betel quid uses) had also affected WBC counts, thus, we used multiple regression analyses to adjust these effects. The blood sample was obtained after at least 8 h fasting. Blood counts were analyzed by a XE-2100 hematology automated analyzer (Sysmex, Wymbush, U.K.) in the Municipal Siaogang Hospital laboratory immediately after blood drawing. This study was approved by the Institutional Review Board of Kaohsiung Medical University Hospital (KMUH-IRB990206) and individual explaining with consent form was done for each subject.

Air quality data were merged to individual blood exam data by the address reported for statistical analysis. For each person, we selected the air pollution data at the blood examination day as “examination day” (ED, or lag 0). Data of the seven days before blood examination were identified as lag 1, lag 2, lag 3, lag 4, lag 5, lag 6, and lag 7 prospectively. Office Excel 2003 (Microsoft, Redmond, Washington, USA) and SPSS for Windows version 20 (IBM, Chicago, IL, USA) were used for descriptive analyses; one-way ANOVA, and multiple linear regressions were employed for inference analyses. Type I error was set as 0.05 with two-tailed.

The multiple linear regression equations are listed as below:
[single pollutant model]
Y= β_0_ + β_1_(age) + β_2_(gender) + β_3_(BMI) + β_4_(cigarette smoking) + β_5_(alcohol consumption) + β_6_(betel quid use) + β_7_(individual pollutant);(1)Y: WBC counts; the individual pollutant: SO_2_, NO_2_, CO, O_3_, PM_10_[multiple pollutants model]Y= β_0_ + β_1_(age) + β_2_(gender) + β_3_(BMI) + β_4_(cigarette smoking) + β_5_(alcohol consumption) + β_6_(betel quid use) + β_7_(SO_2_) + β_8_(NO_2_) + β_9_(CO) + β_10_(O_3_) + β_11_(PM_10_);(2)Y: WBC counts

## 3. Results

Demographic characteristics including age, gender, body mass index (BMI), race, education status as well as smoking, drinking, and betel quid use of enrolled subjects as categorized by district are shown in Table 1. Ten thousand one hundred and forty subjects (4378 males, 43.2%) between the ages of 33–87 years (average 54.2 ± 5.9 years) were recruited. Compared with the other three districts, a higher prevalence of all three unhealthy behaviors (18.4% smoking, 19.6% drinking, and 8.1% betel quid use) were identified among residents in Hsiaokang, where the industrial area is located. WBC counts in these districts are shown in Table 1.

Merging all the measurements of air pollutants in the four districts, Table 2 shows the mean, standard deviation (SD), maximum value, and number of observed sampling for each pollutant in the study periods. The frequencies of measurements of O_3_ and CO were not hourly in all monitoring stations so the observed frequencies of O_3_ and CO were less than those of other pollutants.

Table 3 shows the influences of each air pollutant (including SO_2_, NO_2_, O_3_, CO, and PM_10_) on white blood cells counts as well as differential counts in different time lags (including ED, lag 1 to lag 7, and average of these 8 days) after merge the data of entire districts, and variables of gender, age, body mass index (BMI), smoking, drinking, and betel quid use have been adjusted. Effect of individual air pollutant was identified. Ambient concentration of SO_2_ would decrease monocyte counts. By contrast, NO_2_ as well as O_3_, and PM_10_ would increase counts of total WBC, neutrophils, and monocytes. CO would increase the counts of total WBC and neutrophils, but not monocytes.

Table 4, multiple-pollutants models, shows influences of combined effects of all the studying air pollutants (SO_2_, NO_2_, O_3_, CO, and PM_10_) on white blood cells as well as differential counts in different time lags (including ED, lag 1 to lag 7, and average of these 8 days) with adjustments for gender, age, body mass index (BMI), smoking, drinking, and, betel quid use. Effect of air pollutants on WBC and differential counts on blood exam day was less likely to be identified. Effect of SO_2_ (to decrease the counts), and effect of CO (to increase the counts) on WBC, neutrophils and monocytes were noted in lag 1, lag 2, and lag 3 with statistical significance. Effects of SO_2_ and NO_2_ (to decrease the counts) and effect of CO (to increase the count) on WBC, including neutrophils and monocytes were identified between lag 4 and 7 with statistical significance. The most significant associations and largest magnitudes of pollutant contributions were identified by the value of the average of the 8 days. In WBC model, regression coefficient of SO_2_ was −33.7, NO_2_ was −22.3, and CO was 1431.1.

Table 5 shows the complex effects of average SO_2_, NO_2_, O_3_, CO, and PM_10_ accompanying gender, age, BMI, smoking, drinking, and betel quid use on WBC as well as neutrophils, monocytes, eosinophils, basophils, and lymphocytes. Male gender (regression coefficient 220.7), BMI (regression coefficient 65.3), and cigarette smoking (regression coefficient 642.2) were noted increasing effect on WBC including all the differential counts. Combined effects of five air pollutants with value of average of the 8 days demonstrated that regression coefficient of SO_2_ was −33.7, NO_2_ was −22.3, and CO was 1431.1 in WBC model as well as neutrophils and monocytes.

Table 6 shows the probable changes of WBC counts according to computing the status of air pollution measured in our study (shown in Table 2) and regression coefficients of each pollutant shown in the Table 5. The magnitudes of each air pollutant on WBC could be clearly identified. The largest effect of pollutant would be CO, with an average increase of 858.7 WBC; followed by NO_2_ with an average decrease of 544.1 WBC, and SO_2_ of average decrease of 269.6 WBC.

## 4. Discussion

This study employed a population-based cross-sectional study to examine the relationship between WBC series and short-term ambient concentration (within 8 days) of air pollutants in Kaohsiung. After adjusting the variables of age, gender, BMI, smoking, drinking, and betel quid use, our study revealed the fact that short-term ambient concentrations of SO_2_ and NO_2_ are associated with WBC counts decrease, and short-term ambient concentration of CO is associated with WBC counts increase. In vivo white blood cells change as well as neutrophils and monocytes may trigger or disturb immune systemic or chronic inflammatory process. Both single pollutant as well as multiple-pollutants models are analyzed in this study. Multiple-pollutants model, which consider influences of all the pollutants together, may reflect the real atmosphere status in the air.

Regarding the time-course effects of exposure to air pollution, lag 1 to lag 7 were identified as affective durations to WBC shifting in the present study with statistical significance, and the average value of the 8 days of air pollutants contributed most. The influence of air pollutions on the blood examine day is relatively less. Previous studies obtained a finding that statistical significance of the relationship between WBC and air pollution presented mainly within 5 days lag [31,40,41]. Another study by Liao at al suggested an inconsistent point that lagged exposures (2 or 3 days in their study) were not significantly associated with inflammation variables [42]. A study by Bruske et al. considered the factor of “exposure time”, but no positive conclusion was suggested [43]. Therefore, we take the post exposure day 1 to day 7, a period with impacts most, as short-term exposure.

Sulfur dioxide (SO_2_) was significantly associated with decrease of WBC counts, particularly neutrophils and monocytes, though the effects was weak. SO_2_ is an irritant and colorless environmental pollutant produced by various industrial processes, which showed genotoxic potential and lymphocyte aberration in vivo studies; though there was inconsistency from in vitro studies [18,44,45]. Increased mortality of lung cancer was also identified in SO_2_-exposure subjects [40]. SO_2_ exposure causes health impacts, though its effect on WBC showed inconsistent in vivo results [36,42,45,46]. Inconsistency may present between peripheral tissue and vascular system, and present between long or short duration of inflammation. Liao et al. found a significant curvilinear association between SO_2_ and WBC, this may indicate there is a threshold effect [42]. This curvilinear association model may be able to explain the inconsistent results of our and previous studies. However, further advanced study is warranted to clarify this.

Nitrogen dioxide (NO_2_) can irritate airways and aggravate respiratory disease, particularly asthma, which is link to inflammation mechanism. Under the multiple pollutants model, NO_2_ is negatively associated with WBC count. This result is in line with previous research [32], though some studies find positive association with WBC counts [36]. Most time in our study period, the ambient air quality was under the exposure standard limit of Air Pollution Control Act. Might there is a threshold effect mentioned by previous researchers [42].

A positive association is detected in our study between CO exposure and WBC count increase, especially in neutrophils, monocytes, and lymphocytes. The largest effect of pollutant would be CO, with an average increase of 1431.1 WBC per ppm of CO. The average concentration of CO in our study was 0.6 ppm. Therefore, the probable changes of WBC counts, according to the regression coefficients of CO, demonstrated an 858.7 WBC count increase in our studying subjects. Liao et al. demonstrated a 3 WBC count increase of 0.6 ppm CO exposure, which had a smaller magnitude of CO comparing to our result [42]. Our finding is in line with previous studies [42,47]. Human hemoglobin has much higher affinity to CO than to oxygen, thus CO can seize hemoglobin and decreased oxygen delivery to tissues, result in ischemic status and inflammation response.

In this study, no association between PM exposure and WBC counts is found. Ambient PM is a mixture of components, the compositions vary in different time and different locations. And this brings about different toxicities [19]. Siponen et al. involved 52 ischemic heart disease patients to exam the ability of source-specific fine PM to cause systematic inflammation. The sources of locations were categorized into long-range transport, traffic, biomass combustion, sea salt, and the pulp industry, and it was found that sea salt was not associated with any change of the inflammatory markers [48]. In our research, the participant recruitment locations (districts of Nanzih, Hsiaokang, Cijin, Cianjhen) are nearby the Kaohsiung Harbor, hence sea salts may form part of the PM in these areas. This finding is in line with previous research.

Potential influencing factors including gender, age, BMI, smoking, drinking, and betel quid use have been adjusted in our regression models. Male gender, larger BMI, and smoking have been identified to elevate WBC counts, and the magnitude of smoking is severe in that subjects who smoked would show a WBC increase of 639 counts when compared to non-smokers. Similar suggestions that smokers have higher levels of WBC were identified in the literature [49,50,51]. Increasing WBC per se may explain the inflammatory process in smoking subjects.

Researchers across the world have noted that air pollution is likely associated with adverse health effects, such as cardiovascular disorders, respiratory disorders, malignancy, and others [52]. Possible pathophysiology and processing mechanisms have been studied. Many inflammatory biomarkers, such as interleukin, vascular cell adhesion molecule-1 (VCAM-1), tumor necrosis factor (TNF), intercellular adhesion molecule-1 (ICAM-1), C-reactive protein (CRP) and others have been proposed and studied [53]. The cause-and-effect relationships need further prospective studies, and interactions between biomarkers and specific diseases would be the next-step research topic.

There are some limitations in our research. First, this is an area-level population-based observational study, so extrapolating its applicability to a higher level should be carefully considered. Second, the study participants are volunteers who responded to the invitations, so healthy worker effect bias may be inevitable. Third, the ambient concentrations measurements obtained from air quality monitoring stations are not as accurate as those of individualized personal portable devices, which seem to be able record the exposure in any situation, no matter whether outdoors or indoors. Despite these weaknesses, there are some facts that could compensate partially for this. The main source of indoor air pollutants is from outdoor air [54] and every study district has an air quality monitoring station, therefore the participants in this research should live within 10 kms from the district air pollution monitoring site.

## 5. Conclusions

In this population-based study, we find air pollutants, particularly SO_2_, NO_2_ and CO, are associated with WBC counts. Every unit increase of SO_2_, NO_2_ and CO, is associated with WBC count changes of −33.7, −22.3, and 1431.1, respectively. These would imply that air pollution has an impact on systematic inflammation responses. Our research finding is in line with previous studies, and supports the postulated link between air pollution and inflammation, which is a common pathway sparking off human diseases. However, this study is on a local area scale, and further prospective, larger-scale (ex. nationwide) studies are recommended. Variables associated with WBC count, such as exercise, lipid profiles, or another inflammation-related biomarkers could be considered in future research.

## Figures and Tables

**Figure 1 ijerph-18-02370-f001:**
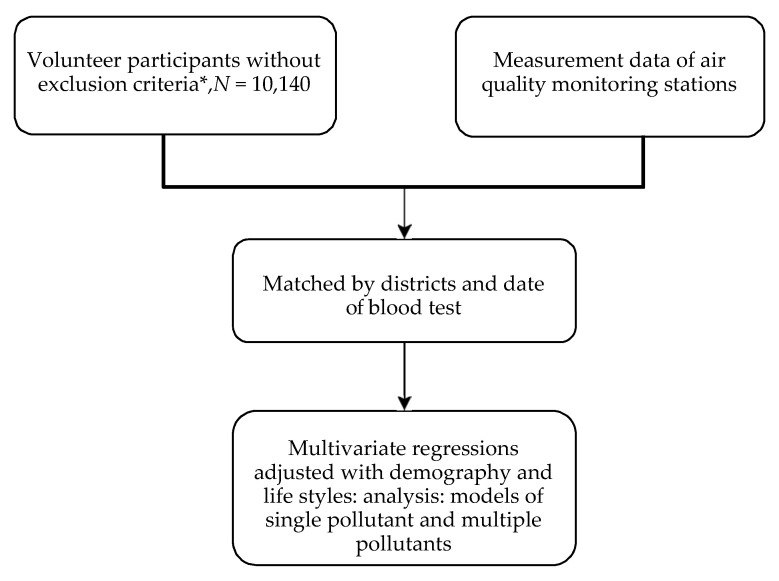
Brief of method steps. *, exclusion criteria: pregnancy, current malignancy diseases, history of auto-immune diseases, possible infectious conditions (such as common cold) or fever, and age < 30 years-old.

**Table 1 ijerph-18-02370-t001:** Demographic characteristics of participants and WBC counts in study districts.

	Districts	Nanzih (*n* = 2466)		Hsiaokang (*n* = 2508)		Cijin (*n* = 335)		Cianjhen (*n* = 4831)	
Variables									
Age, year (mean ± SD)		54.7 ± 5.5		54.2 ± 5.9		54.3 ± 7.3		54.1 ± 6.1	
Gender, male		1094 (44.4%)		1146 (45.7%)		166 (49.6%)		1972 (40.8%)	
BMI, kg/m^2^ (mean ± SD)		24.6 ± 4.0		24.6 ± 3.4		25.5 ± 3.8		24.8 ± 3.9	
Education level	College/Graduate	354	(14.4%)	221	(8.8%)	24	(7.2%)	732	(15.2%)
	High school	984	(39.9%)	940	(37.5%)	107	(31.9%)	2019	(41.8%)
	Elementary school	837	(33.9%)	927	(37.0%)	126	(37.6%)	1532	(31.7%)
	Illiteracy	266	(10.8%)	405	(16.2%)	75	(22.4%)	512	(10.6%)
Cigarette smoking	current	385	(15.6%)	461	(18.4%)	51	(15.2%)	612	(12.7%)
	former	130	(5.3%)	63	(2.5%)	26	(7.8%)	303	(6.3%)
	never	1926	(78.1%)	1968	(78.5%)	250	(74.6%)	3780	(78.2%)
Alcohol consumption	yes	383	(15.5%)	492	(19.6%)	41	(12.2%)	502	(10.4%)
	no	2046	(83.0%)	1998	(79.7%)	290	(86.6%)	4265	(88.3%)
Betel quid use	yes	137	(5.6%)	202	(8.1%)	18	(5.4%)	300	(6.2%)
	no	2301	(93.3%)	2280	(90.9%)	314	(93.7%)	4481	(92.8%)
WBC count/μL, (mean ± SD)	total	5915.8 ± 1458.0		5726.9 ± 1426.4		6290.3 ± 1538.9		5982.2 ± 5982.2	
	neutrophile	3161.2 ± 1108.3		3092.7 ± 1081.7		3511.7 ± 1203.9		3327.4 ± 1148.4	
	lymphocyte	2179.8 ± 633.2		2104.0 ± 591.6		2201.7 ± 615.8		2111.9 ± 621.1	
	monocyte	381.0 ± 135.0		345.0 ± 125.1		381.2 ± 130.3		356.2 ± 124.8	
	eosinophile	156.2 ± 135.8		153.8 ± 130.9		162.3 ± 123.3		148.2 ± 128.1	
	basophil	32.5 ± 22.1		30.6 ± 20.5		33.5 ± 21.6		30.7 ± 20.8	
Air pollutants (mean ± SD)	SO_2_ (ppb)	6.2 ± 3.1		9.6 ± 5.1		11.6 ± 3		8.4 ± 1.9	
	NO_2_ (ppb)	19.5 ± 4.4		18.8 ± 5.1		27.5 ± 15.6		31.3 ± 12.1	
	O_3_ (ppb)	29.9 ± 10.3		24.5 ± 11.1		27.1 ± 7.1		*	
	CO (ppm)	0.6 ± 0.1		0.6 ± 0.2		1.1 ± 0.4		*	
	PM_10_ (μg/m^3^)	61 ± 18.3		49 ± 17.2		67.2 ± 23.8		87.7 ± 33.4	

*, no apparatus measurement.

**Table 2 ijerph-18-02370-t002:** Status of environmental pollutants which our subjects exposed to.

Pollutant	*n*	Min	Mean (SD)	Q1	Q3	Max
SO_2_ (ppb)	9767	0.40	8.25 (3.61)	5.51	10.25	44.00
NO_2_ (ppb)	9767	1.20	25.00 (11.11)	16.21	35.61	115.60
O_3_ (ppb)	5286	5.00	27.17 (10.83)	17.46	38.54	62.60
CO (ppm)	5286	0.23	0.61 (0.21)	0.46	0.72	2.20
PM_10_ (μg/m^3^)	9664	10.92	70.22 (31.07)	42.56	92.61	289.0

Q1, 25 percentiles; Q3, 75 percentiles; Min, minimum; Max, maximum value; *n*, observed number.

**Table 3 ijerph-18-02370-t003:** Single-pollutant model: the associations between common air pollutants and white blood cells after adjusting gender, age, BMI, smoking, drinking, and betel quid chewing.

Pollutants	White Blood Cells		Neutrophil Counts		Monocyte Counts		Eosinophil Counts		Basophil Counts		Lymphocyte Counts	
	β	(SE)	β	(SE)	β	(SE)	β	(SE)	β	(SE)	β	(SE)
SO_2_ (ED)	4.12	(2.87)	2.12	(2.24)	0.36	(0.24)	0.21	(0.26)	−0.70	(1.39)	1.48	(1.22)
SO_2_ (lag 1)	−3.98	(2.64)	−0.18	(2.06)	−1.21	(0.22) **	−0.22	(0.24)	−0.47	(1.28)	−2.40	(1.12)
SO_2_ (lag 2)	−2.81	(2.77)	−1.83	(2.16)	−0.46	(0.24)	−0.26	(0.25)	0.50	(1.34)	−0.32	(1.17)
SO_2_ (lag 3)	−1.25	(2.71)	1.65	(2.12)	−1.26	(0.23) **	−0.61	(0.25) *	0.68	(1.33)	−1.37	(1.15)
SO_2_ (lag 4)	−3.33	(2.74)	−1.54	(2.15)	−1.18	(0.23) **	−0.84	(0.25) **	−1.14	(1.34)	0.38	(1.16)
SO_2_ (lag 5)	−4.18	(2.69)	−0.98	(2.12)	−1.40	(0.23) **	−0.54	(0.24)	−0.91	(1.33)	−0.91	(1.14)
SO_2_ (lag 6)	−6.87	(2.60) **	−2.99	(2.04)	−1.38	(0.22) **	−0.48	(0.23)	−0.44	(1.29)	−1.46	(1.10)
SO_2_ (lag 7)	−3.87	(2.54)	−3.22	(2.00)	−0.91	(0.22) **	−0.09	(0.23)	−0.34	(1.26)	0.37	(1.07)
a_SO_2_	−6.36	(4.07)	−1.59	(3.19)	−2.18	(0.35) **	−0.83	(0.37) *	−0.81	(1.96)	−1.56	(1.73)
NO_2_ (ED)	7.72	(1.14) **	5.15	(0.89) **	0.43	(0.10) **	0.20	(0.10)	0.47	(0.56)	2.01	(0.48) **
NO_2_ (lag 1)	7.56	(1.14) **	5.85	(0.89) **	0.41	(0.10) **	0.24	(0.10) *	0.50	(0.55)	1.12	(0.48) *
NO_2_ (lag 2)	8.67	(1.22) **	6.46	(0.96) **	0.42	(0.10) **	0.12	(0.11)	0.98	(0.60)	1.71	(0.52) **
NO_2_ (lag 3)	8.17	(1.28) **	6.98	(1.00) **	0.19	(0.11)	0.03	(0.12)	0.62	(0.62)	0.92	(0.54)
NO_2_ (lag 4)	9.96	(1.20) **	7.34	(0.94) **	0.48	(0.10) **	0.09	(0.11)	0.92	(0.58)	2.17	(0.51) **
NO_2_ (lag 5)	8.80	(1.21) **	7.40	(0.94) **	0.34	(0.10) **	0.05	(0.11)	0.83	(0.58)	1.06	(0.51) *
NO_2_ (lag 6)	7.96	(1.16) **	6.64	(0.90) **	0.39	(0.10) **	0.11	(0.10)	0.51	(0.56)	0.94	(0.49)
NO_2_ (lag 7)	7.61	(1.14) **	6.07	(0.89) **	0.43	(0.10) **	0.05	(0.10)	0.46	(0.56)	1.12	(0.48) *
a_NO_2_	11.00	(1.33) **	8.45	(1.04) *	0.52	(0.11) **	0.16	(0.12)	0.82	(0.64)	1.95	(0.57) **
O_3_ (ED)	5.47	(1.45) **	2.79	(1.13) *	0.52	(0.13) **	0.33	(0.13) *	0.03	(0.02)	1.87	(0.62) **
O_3_ (lag 1)	4.55	(1.53) **	1.46	(1.20)	0.27	(0.14)	0.35	(0.14) *	0.00	(0.02)	2.61	(0.66) **
O_3_ (lag 2)	5.18	(1.48) **	2.56	(1.15) *	0.31	(0.13) *	0.33	(0.14) *	0.03	(0.02)	2.10	(0.63) **
O_3_ (lag 3)	5.57	(1.47) **	2.44	(1.15) *	0.62	(0.13) **	0.46	(0.14) **	0.04	(0.02)	2.12	(0.63) **
O_3_ (lag 4)	5.00	(1.38) **	2.37	(1.08) *	0.66	(0.12) **	0.46	(0.13) **	0.03	(0.02)	1.45	(0.59) *
O_3_ (lag 5)	6.22	(1.37) **	3.18	(1.07) **	0.85	(0.12) **	0.45	(0.13) **	0.06	(0.02) **	1.72	(0.59) **
O_3_ (lag 6)	4.97	(1.42) **	2.71	(1.11) *	0.75	(0.13) **	0.36	(0.13) **	0.06	(0.02) **	1.16	(0.61)
O_3_ (lag 7)	3.42	(1.42) *	2.78	(1.11) *	0.38	(0.13) **	0.30	(0.13) *	0.06	(0.02) **	−0.03	(0.61)
a_O_3_	8.08	(1.82) **	4.04	(1.42) **	0.89	(0.16) **	0.61	(0.17) **	0.06	(0.03) *	2.55	(0.78) **
CO (ED)	337.07	(77.93) **	240.27	(60.71) **	13.13	(6.91)	16.12	(7.21) *	2.64	(1.18) *	70.50	(33.34) *
CO (lag 1)	362.17	(77.34) **	289.18	(60.25) **	11.07	(6.87)	10.45	(7.18)	2.27	(1.17)	57.05	(33.05)
CO (lag 2)	295.36	(78.85) **	162.97	(61.58) **	16.24	(7.01) *	12.95	(7.12)	2.04	(1.17)	107.69	(33.74) **
CO (lag 3)	343.45	(79.59) **	269.17	(62.06) **	6.06	(7.08)	4.65	(7.19)	3.01	(1.19) *	66.02	(34.13)
CO (lag 4)	324.90	(85.52) **	238.86	(66.71) **	8.56	(7.58)	4.27	(7.70)	3.02	(1.27) *	79.92	(36.71) *
CO (lag 5)	242.71	(87.89) **	210.08	(68.43) **	−0.66	(7.79)	11.59	(8.13)	3.22	(1.33) *	24.05	(37.60)
CO (lag 6)	266.09	(84.07) **	275.30	(65.49) **	−8.49	(7.45)	10.27	(7.78)	2.88	(1.27) *	−5.43	(35.98)
CO (lag 7)	264.40	(84.40) **	219.94	(65.89) **	−1.89	(7.48)	7.92	(7.81)	2.50	(1.28)	41.44	(36.12)
a_CO	424.80	(95.84) **	329.63	(74.69) **	8.44	(8.50)	13.64	(8.87)	3.76	(1.45) *	78.56	(41.04)
PM_10_ (ED)	3.33	(0.44) **	2.34	(0.35) **	0.22	(0.04) **	0.04	(0.04)	0.24	(0.22)	0.77	(0.19) **
PM_10_ (lag 1)	3.30	(0.44) **	2.59	(0.34) **	0.13	(0.04) **	0.03	(0.04)	0.30	(0.22)	0.59	(0.19) **
PM_10_ (lag 2)	3.39	(0.44) **	2.44	(0.35) **	0.20	(0.04) **	0.02	(0.04)	0.60	(0.22) **	0.76	(0.19) **
PM_10_ (lag 3)	3.34	(0.43) **	2.43	(0.33) **	0.19	(0.04) **	0.05	(0.04)	0.37	(0.21)	0.66	(0.18) **
PM_10_ (lag 4)	3.76	(0.41) **	2.66	(0.32) **	0.26	(0.04) **	0.07	(0.04)	0.47	(0.20) *	0.75	(0.18) **
PM_10_ (lag 5)	3.86	(0.45) **	2.59	(0.35) **	0.29	(0.04) **	0.09	(0.04) *	0.13	(0.22)	0.89	(0.19) **
PM_10_ (lag 6)	3.01	(0.40) **	2.29	(0.31) **	0.21	(0.03) **	0.07	(0.04)	0.05	(0.20)	0.48	(0.17) **
PM_10_ (lag 7)	2.95	(0.44) **	2.40	(0.34) **	0.14	(0.04) **	0.04	(0.04)	0.37	(0.22)	0.41	(0.19) **
a_PM_10_	4.30	(0.48) **	3.12	(0.37) **	0.26	(0.04) **	0.05	(0.04)	0.37	(0.23)	0.90	(0.20) **

β, regression coefficients; SE, standard errors; ** *p* < 0.01; * *p* < 0.05; ED, examination day or lag 0; a, average of the 8 days.

**Table 4 ijerph-18-02370-t004:** Multiple-pollutant model: the associations between common air pollutants and white blood cells after adjusting gender, age, BMI, smoking, drinking, and betel quid chewing.

Lag Days		White Blood Cells		Neutrophil Counts		Monocyte Counts		Eosinophil Counts		Basophil Counts		Lymphocyte Counts	
		β	(SE)	β	(SE)	β	(SE)	β	(SE)	β	(SE)	β	(SE)
ED	SO_2_	0.6	(4.7)	−3.2	(3.6)	1.1	(0.4)	0.3	(0.4)	−0.1	(0.1)	2.4	(2.0)
	NO_2_	1.4	(3.8)	−0.7	(3.0)	−0.1	(0.3)	1.1	(0.4) *	0.0	(0.1)	1.1	(1.6)
	O_3_	5.3	(2.7)	2.8	(2.1)	0.6	(0.2)	0.4	(0.3)	0.0	(0.0)	1.4	(1.2)
	CO	288.4	(161.9)	378.0	(125.7) *	−17.7	(14.4)	−18.2	(15.1)	2.4	(2.4)	−51.7	(69.1)
	PM_10_	−1.4	(1.9)	−1.8	(1.5)	0.2	(0.2)	−0.1	(0.2)	0.0	(0.0)	0.4	(0.8)
Lag 1	SO_2_	−13.0	(4.7) **	−7.8	(3.7)	−2.5	(0.4) **	−0.3	(0.4)	−0.2	(0.1)	−2.2	(2.0)
	NO_2_	5.5	(3.9)	3.7	(3.0)	0.9	(0.3) *	1.2	(0.4) *	0.1	(0.1)	−0.6	(1.7)
	O_3_	−1.3	(3.0)	−3.4	(2.3)	−0.4	(0.3)	0.5	(0.3)	−0.1	(0.0)	1.9	(1.3)
	CO	436.0	(174.2) *	340.4	(135.6) **	43.0	(15.4) **	−8.8	(16.2)	3.6	(2.6)	62.9	(74.4)
	PM_10_	−0.8	(1.7)	0.0	(1.3)	−0.3	(0.2)	−0.3	(0.2)	0.0	(0.0)	0.0	(0.7)
Lag 2	SO_2_	−17.0	(4.9) **	−9.2	(3.8) *	−2.1	(0.4) **	−0.8	(0.4)	−0.1	(0.1)	−4.6	(2.1) *
	NO_2_	2.0	(3.3)	2.9	(2.6)	0.0	(0.3)	0.5	(0.3)	0.1	(0.0)	−1.7	(1.4)
	O_3_	3.0	(2.8)	3.3	(2.2)	−0.5	(0.2)	0.1	(0.2)	0.0	(0.0)	0.2	(1.2)
	CO	623.2	(164.3) **	390.6	(128.2) **	44.1	(14.6) **	9.1	(14.8)	2.5	(2.4)	182.2	(70.4) *
	PM_10_	−3.5	(1.8)	−3.9	(1.4) **	0.1	(0.2)	0.0	(0.2)	0.0	(0.0)	0.4	(0.8)
Lag 3	SO_2_	−13.9	(4.5) **	−5.3	(3.5)	−2.5	(0.4) **	−1.0	(0.4)	0.0	(0.1)	−4.7	(1.9) *
	NO_2_	−4.0	(4.0)	−0.5	(3.1)	−0.4	(0.4)	0.2	(0.4)	0.0	(0.1)	−3.5	(1.7) *
	O_3_	2.8	(2.5)	2.7	(1.9)	0.0	(0.2)	0.4	(0.2)	0.0	(0.0)	−0.2	(1.1)
	CO	690.7	(151.6) **	462.9	(118.3) **	49.1	(13.4) **	15.4	(13.7)	1.6	(2.3)	160.1	(65.1) *
	PM_10_	−1.7	(1.6)	−2.5	(1.2)	0.0	(0.1)	−0.2	(0.1)	0.0	(0.0)	1.0	(0.7)
Lag 4	SO_2_	−14.3	(4.4) **	−11.5	(3.4) **	−1.8	(0.4) **	−0.7	(0.4)	−0.1	(0.1)	−0.4	(1.9)
	NO_2_	−13.0	(4.6) **	−11.1	(3.6) **	−1.5	(0.4) **	0.0	(0.4)	0.0	(0.1)	−0.4	(2.0)
	O_3_	−0.9	(2.6)	−1.2	(2.0)	−0.3	(0.2)	0.3	(0.2)	−0.1	(0.0)	0.4	(1.1)
	CO	718.2	(148.5) **	614.9	(116.4) **	40.2	(13.1) **	5.3	(13.4)	3.0	(2.2)	70.3	(63.6)
	PM_10_	1.5	(1.8)	0.3	(1.4)	0.5	(0.2) *	0.0	(0.2)	0.1	(0.0)	0.6	(0.8)
Lag 5	SO_2_	−10.6	(4.2) *	−8.5	(3.3) **	−1.8	(0.4) **	−0.6	(0.4)	−0.1	(0.1)	−0.1	(1.8)
	NO_2_	−10.2	(4.0) *	−3.7	(3.2)	−1.2	(0.4) **	0.0	(0.4)	0.1	(0.1)	−5.6	(1.7) **
	O_3_	2.7	(2.5)	3.4	(1.9)	0.0	(0.2)	0.0	(0.2)	0.0	(0.0)	−0.7	(1.1)
	CO	533.3	(156.7) **	470.4	(122.7) **	27.7	(13.8) *	4.9	(14.5)	2.3	(2.4)	48.6	(67.0)
	PM_10_	0.8	(1.9)	−2.5	(1.5)	0.4	(0.2) **	0.2	(0.2)	0.0	(0.0)	2.5	(0.8) **
Lag 6	SO_2_	−16.7	(4.0) **	−12.3	(3.2) **	−1.6	(0.4) **	−0.8	(0.4) *	−0.2	(0.1)	−1.6	(1.7)
	NO_2_	−17.0	(4.4) **	−9.3	(3.4) **	−1.7	(0.4) **	−0.9	(0.4) *	0.0	(0.1)	−4.8	(1.9) *
	O_3_	0.0	(2.4)	−1.4	(1.8)	0.3	(0.2)	0.1	(0.2)	0.0	(0.0)	1.4	(1.0)
	CO	761.4	(155.3) **	583.4	(120.8) **	32.4	(13.8) **	33.4	(14.5) *	5.3	(2.4)	110.3	(66.6)
	PM_10_	2.5	(1.8)	1.7	(1.4)	0.4	(0.2)	0.2	(0.2)	0.0	(0.0)	0.0	(0.8)
Lag 7	SO_2_	−15.3	(4.1) **	−10.8	(3.2) *	−1.6	(0.4) **	−0.4	(0.4)	−0.2	(0.1)	−2.2	(1.8)
	NO_2_	−16.6	(4.1) **	−10.2	(3.2) *	−0.9	(0.4) *	−1.0	(0.4) *	−0.1	(0.1)	−4.1	(1.8) *
	O_3_	−3.9	(2.3)	−2.6	(1.8)	−0.1	(0.2)	0.2	(0.2)	0.0	(0.0)	−1.2	(1.0)
	CO	760.0	(169.3) **	538.0	(132.0) **	36.5	(15.0) *	28.6	(15.7)	5.1	(2.6)	147.0	(72.8) *
	PM_10_	3.5	(1.8)	2.3	(1.4)	0.2	(0.2)	0.1	(0.2)	0.0	(0.0)	0.8	(0.8)
a_SO_2_		−33.7	(7.3) **	−23.9	(5.7) **	−4.4	(0.6) **	−1.3	(0.7)	−0.4	(0.1) **	−3.8	(3.1)
a_NO_2_		−22.3	(7.0) **	−10.2	(5.5)	−2.2	(0.6) **	0.5	(0.6)	0.1	(0.1)	−10.7	(3.0) **
a_O_3_		0.2	(4.6)	0.9	(3.6)	−0.3	(0.4)	0.5	(0.4)	−0.1	(0.1)	−0.7	(2.0)
a_CO		1431.1	(258.2) **	1061.9	(201.7) **	105.0	(22.8) **	22.8	(24.0)	6.6	(3.9)	243.4	(110.8) *
a_PM_10_		−1.3	(3.3)	−3.9	(2.5)	0.2	(0.3)	−0.3	(0.3)	0.0	(0.0)	2.7	(1.4)	

β, regression coefficients; SE, standard errors; ** p* < 0.05; ** *p* < 0.01; ED, examination day or lag0; a, average of the 8 day.

**Table 5 ijerph-18-02370-t005:** Multiple-pollutants model: the associations of combined effects between common air pollutants and white blood cell series.

	White Blood Cells		Neutrophil Counts		Monocyte Counts		Eosinophil Counts		Basophil Counts		Lymphocyte Counts	
	β	(SE)	β	(SE)	β	(SE)	β	(SE)	β	(SE)	β	(SE)
Gender (male)	220.7	(46.1) **	116.2	(36.0) **	57.92	(4.1) **	36.76	(4.3) **	1.76	(0.7) *	6.66	−19.8
Age	4	−3.5	−0.49	−2.7	1.17	(0.3) **	−0.09	−0.3	0.06	−0.1	3.12	−1.5
BMI	65.3	(5.4) **	29.54	(4.2) **	3.83	(0.5) **	2.4	(0.5) **	0.38	(0.1) **	28.67	(2.3) **
Cigarette smoking	642.2	(61.4) **	321.48	(48.0) **	47.33	(5.4) **	33.52	(5.7) **	4.39	(0.9) **	239.21	(26.4) **
Alcohol drinking	−79.8	−59.3	−64.71	−46.3	−7.93	−5.2	−8.52	−5.5	0.72	−0.9	1.01	−25.4
Betel quid use	−14.9	−123.1	−22.57	−95.9	−7.47	−10.9	26.61	(11.4) *	0.7	−1.9	−19.22	−52.8
a_SO_2_	−33.7	(7.3) **	−23.86	(5.7) **	−4.35	(0.6) **	−1.27	−0.7	−0.37	(0.1) **	−3.8	−3.1
a_NO_2_	−22.3	(7.0) **	−10.15	−5.5	−2.19	(0.6) **	0.47	−0.6	0.07	−0.1	−10.67	(3.0) **
a_O_3_	0.2	−4.5	0.89	−3.6	−0.27	−0.4	0.55	−0.4	−0.11	−0.1	−0.69	−2
a_CO	1431.1	(258.3) **	1061.91	(201.7) **	105.01	(22.8) **	22.82	−24	6.56	−3.9	243.39	(110.8) *
a_PM_10_	−1.3	−3.23	−3.89	−2.5	0.18	−0.3	−0.32	−0.3	0.02	−0.1	2.73	−1.4

β, regression coefficients; SE, standard errors; * *p* < 0.05; ** *p* < 0.01; a, average of the 8 days.

**Table 6 ijerph-18-02370-t006:** Probable changes of WBC count influence by air pollutions in Kaohsiung City.

Pollutant	Interquartile in Air (Q3−Q1)	Adjusted Regression Coefficients (SE) of WBC Counts	WBC Changes from Q1 to Q3
SO_2_	4.74 ppb	−33.7 (7.3) **	−159.7
NO_2_	19.4 ppb	−22.3 (7.0) **	−432.6
O_3_	21.08 ppb	0.2 (4.5)	4.2
CO	0.25 ppm	1431.1 (258.3) **	357.8
PM_10_	50.05 μg/m^3^	−1.3 (3.23)	−65.1

Q1, 25 percentiles; Q3, 75 percentiles; ** *p* < 0.01.

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
