# Peer review of "The Association of White Blood Cells and Air Pollutants—A Population-Based Study"

_ijerph, 2021, doi:10.3390/ijerph18052370_

Round 1

Reviewer 1 Report

This is a study which evaluates the effects of short-term exposure   to air pollutant on overall counts of WBC. Authors suggest that  short-term exposure to air pollutants, may be related to increased WBC counts, thus explaining how air pollutants  would impact on  immune system. However major questions must be addressed: 

  1. It is not clear in the abstract section the impact of air pollutants in health conditions. 
  2. The methods section must be improved:  2.1-It is not clear what exclusion criteria were adopted in the present study. For example, some chronic inflammatory diseases could alter the WBC counting. These participants were excluded from the study? 2.2- Also, it is not clear what is the control group of the study, more information is needed in the manuscript. 2.3-Overall, the methods section should be improved. What does it mean the N in table 2? 2.4-Table descriptions are poor and must be improved.
  3. The analysis  of circulating inflammatory mediators (biomarkers) are very strongly recommended  to provide a more convincing and robust idea about the effects of air pollutants over inflammatory processes.

  4. Authors point that a positive association between CO exposure and neutrophils, monocytes, and lymphocyte counts, however there is still a lacking of absolute count numbers of these cells to evaluate the magnitude of such effect on the immune system.

  5. Discussion section must be improved. Authors must at least discuss the impact of the increase and/or decrease of WBC on health and its contribution several diseases. For example, in the introduction sectins authors point that inflammatory process provoked by air pollution have been suggested by previous researches. Whats are these processes? How changing in WBC counting affects inflammation? Which is the contribution of the different WBC types on these processes?

Author Response

  1. Thank for reviewer’s advice. In spite of the limitation of 200 words maximum, we have revised the abstract paragraph to state more why we research into the impact of air pollution on WBC counts.
  2. The exclusion criteria have been added to articles, (line 92-94). They included pregnancy, current malignancy disease, history of auto-immune disease, and other infectious condition (like as common cold) or fever on the exam day, and age < 30.
  3. About the control group, this is the limitation of our study, owing to population-bases observational survey. However, we have considered of the exposure effect of daily lag of air pollutants.
  4. The methods section has been improved. In table 2, “N” is the observed numbers.
  5. Tables descriptions have been revised to improve.
  6. About the recommendation of analysis of inflammatory biomarkers, this is one of the limitations of our study. Further researches on the inflammation related biomarker are promising and needed.  And we put this information in our limitation section and recommendation of conclusion paragraph. 
  7. The WBC counts have been added in table 1.
  8. Discussion section has been improved. We have added more information about inflammation biomarkers and adverse effect of air pollution on health.

Reviewer 2 Report

Review of the article “The Association of Wide Blood Cells and Air Pollutants A Population-Based Study

Shortcomings of the article:

The purpose of the study is unclear.

The work cites old literature and only about 18 percent i.e. 7 articles out of 39 articles not older than 5 years. There is no point in quoting old literature. The last source of literature is not numbered. The literature review was done very dismissively. Each source should be discussed and the importance of its citation explained. I think the introduction should be rewritten to include new literature sources no older than 5 years and a serious analysis of the literature presented should be done.

A clear research methodology must be provided on how the results are obtained and how reliable they are.

Needs add information about the research application possibilities in practice.

In the conclusions must clearly show what problems the researchers have solved and which results are better than the results of other researches. Rewrite the conclusions summarizing the steps followed for the development of the results and highlighting the results obtained.

The article is not a report now submitted work looks like a report.

The article needs to be substantially redesigned to be suitable for publication.

Author Response

  1. Thank for reviewer's advice. We have added a description of  study purpose in the abstract and in the introduction section.
  2. The introduction and method sections have been revised to improve.
  3. About citing old literature, we have included more citation of recent studies. Although some of the citations are old, but seem not outdate, and some of their opinions/findings echo our research, so we still keep the citations.
  4. About research application possibilities in practice, we have added the information about current practice of our government, Environment Protection Administration (EPA) .
  5. The conclusion paragraph has been rewritten to improve.

Reviewer 3 Report

Comments

In this manuscript, Hung and colleagues studied the association of white blood cells and air pollutants using environmental pollutant data from air quality monitoring stations and the blood counts data generated by hematology automated analyzer. They found that short-term exposure to air pollutants, especially CO and SO2, related to white blood cell count changes. The study is very interesting and the manuscript is well presented in general, while the authors should address the following concerns before the study can be considerate for publication.

  1. Information of “Methods” is not adequate. Please explain in “method” how “variables of gender, age,.. have been adjusted” (lines 161-162 and 171-172). Please also provide the detailed information for the analysis of “multiple-pollutant model”.
  2. Please indicate the “units” of WBC counts of Table 1 and also indicate whether the units for while blood counts in the manuscript are the same or not.
  3. Line 153-156, please double check if the numbers match the numbers in Table 2.
  4. It will be better if the authors can change Table 2 to Figure so that readers will get more information about the pollutants, including the highest, average, lowest, duration et al.

Author Response

Review Report (Reviewer 3)

In this manuscript, Hung and colleagues studied the association of white blood cells and air pollutants using environmental pollutant data from air quality monitoring stations and the blood counts data generated by hematology automated analyzer. They found that short-term exposure to air pollutants, especially CO and SO2, related to white blood cell count changes. The study is very interesting and the manuscript is well presented in general, while the authors should address the following concerns before the study can be considerate for publication.

Information of “Methods” is not adequate. Please explain in “method” how “variables of gender, age,... have been adjusted” (lines 161-162 and 171-172). Please also provide the detailed information for the analysis of “multiple-pollutant model”.

Response: Thanks for the comment. We have added the point in the section of method, page 4. In brief, the multivariate regression equations are listed as below:

  • [single pollutant model]
    Y= β0 + β1(age) + β2(gender) + β3(education) + β4(cigarette smoking) + β5(alcohol consumption) + β6(betel quid use) + β7(individual pollutant);
    Y: WBC counts; the individual pollutant: SO2, NO2, CO, O3, PM10
  • [multiple pollutants model]
    Y= β0 + β1(age) + β2(gender) + β3(education) + β4(cigarette smoking) + β5(alcohol consumption) + β6(betel quid use) + β7(SO2) + β8(NO2) + β9(CO) + β10(O3) + β11(PM10);
    Y: WBC counts

Please indicate the “units” of WBC counts of Table 1 and also indicate whether the units for while blood counts in the manuscript are the same or not.

Response: Thanks for the comment. The “units” of WBC, count/μL, has been added.

Line 153-156, please double check if the numbers match the numbers in Table 2.

Response: Thanks for the comment. Line 153-156 and table 2 were corrected, and add Maximum and minimum.

It will be better if the authors can change Table 2 to Figure so that readers will get more information about the pollutants, including the highest, average, lowest, duration et al.

Response: Thanks for the comment, however, because the units of pollutants are different and the number scale is wide, we think table style can show the picture clearly and easily.

Round 2

Reviewer 1 Report (none)

Reviewer 2 Report

The authors of the article did not take the submitted comments seriously enough. The corrections made need to be supplemented with special attention to the literature review, conclusions and the list of cited literature. The literature review is supplemented by only 3 and not the most recent articles.

Author Response

    In this revision, after reviewing more updated references, we have rewritten the section of introduction and have made many modifications for improvement in the section of discussion. Thank you very much.

Reviewer 3 Report

The authors have only partially addressed most of the comments and suggestions. There are still several requirements that have not been answered satisfactorily.

For example, the introduction section has not been modified according to the requests and suggestions. Furthermore, the incorporation of more recent references has not been considered. Several works concerning the effect of particulate matter, heavy metals or persistent organic pollutants on the leukocyte populations during the last decade.

In Methods, the calculation of the sample size and the requirements regarding the effect of  the environmental conditions and lifestyle have not either been reported or included as limitations of the study.

In Results, the recommendations suggested have not been taken into account and the discussion of the parameters requested has not been addressed.

The conclusion section has not improved significantly.

Author Response

About sample size, it was depended on the population who agreed to participate our research. We have made our best efforts to collect over ten thousand participants. We have only some lifestyle variables, such as smoking, alcohol, betel quid use, and they were included in the model of multiple regression as covariates (table 5). Thank you very much.

        After another reviewing more updated references, the sections of discussion and conclusion have been modified for improvement on the basis of result section. Thank you very much.

Academic Editor: Reject and encourage resubmission

In the introduction, the work's purpose is not clear: whether there is a correlation between white blood cells and air pollution and why the authors wanted to address this issue.

Then, as suggested by reviewers 2 and 3, the introduction and discussion must be rewritten based on the new literature on this topic from 2015. Updating the literature does not mean adding 3 more articles in the citations. Several works concerning the effect of particulate matter, heavy metals, or persistent organic pollutants on the leukocyte populations during the last decade.

(for es. Chen, et al  Size-fractionated Particulate Air Pollution and Circulating Biomarkers of Inflammation, Coagulation, and Vasoconstriction in a Panel of Young Adults Epidemiology 2015; Matthew A Stiegel et al Inflammatory Cytokines and White Blood Cell Counts Response to Environmental Levels of Diesel Exhaust and Ozone Inhalation Exposures PLoS One  2016 Apr 8;11(4):e0152458. doi: 10.1371/journal.pone.0152458; Xu Gao et al., Impacts of air pollution, temperature, and relative humidity on leukocyte distribution: An epigenetic perspective Environ Int. 2019 May;126:395-405. doi: 10.1016/j.envint.2019.02.053.)

Second, it was not explained by the authors the reasons for considering the period studied and the implications of studying in the environmental conditions occurring in 2003-2004 (15 years ago!) In the method, the authors must explain in detail the statistical methods used and why they used them.

Finally, in the feeble discussion, the authors should make hypotheses, comparing with the literature, of why an increase in CO and SO2 may be related to an increase and decrease of WBC counts, respectively.

Author Response

Dear editors:

Enclosed is a revised manuscript entitled "The Association of White Blood Cells and Air Pollutants– A Population-Based Study.", which we are submitting to the International Journal of Environmental Research and Public Health. The original research is not currently under consideration by any other journals. The study protocol was approved by the institutional review board at Kaohsiung Medical University Hospital. All participants gave informed written consents after detailed explanation of the study, and then procedure was done. All authors have read the manuscript and agree to submit, as well as disclosed any potential competing financial interests.
To meet the suggestions of reviewers, we have re-written the introduction section, made a major modification for the discussion section, and updated most of the references in this revised edition.

The data collection had been kept in Department of Health, Kaohsiung City Government since the investigation was completed, and city government just opened the data years ago. And this is the reason we re-do the analysis so late.

We thought that the International Journal of Environmental Research and Public Health readers (including both practitioners and researchers) might find it helpful as an alternative approach to providing more comprehensive investigations, and practice in environmental and occupational health.

Please kindly let us know if the contents could be of interest to the International Journal of Environmental Research and Public Health. Thanks in advance.

Best Regards,

Round 3

Reviewer 2

The Association of White Blood Cells and Air Pollutants– A Population-Based Study
Authors: Shih-Chiang Hung, Hsiao-Yuan Cheng, Chen-Cheng Yang, Chia-I Lin, Chi-Kung Ho, Wen-Huei Lee, Fu-Jen Cheng, Chao-Jui Li, Hung-Yi Chuang

The methodic of researches is very generalized. It lacks the specificity of clarity for example how selections was made and so on.

The researches methodology could be presented in a different form e.g. drawn diagram or somehow similar.

There is no information on how the results were obtained or how the calculations were performed. What is the reliability of the results. The theoretical-numerical part of the researches needs to be presented.

The conclusions should be brief, but they should include: what specific results have been obtained from the studies and compared with the results of other researchers. The benefits of your research are explained. The Conclusions presented in this article are of no scientific benefits.

Author response

  • Thank for reviewer's advise, we have revised the manuscript to improve.
  • In spite of the limitation of 200 words maximum, we has added the age range information in the abstract paragraph.
  • About citing more recent references, we have included more citation of recent studies. Although some of the citations are old, but seem not outdate, and some of their opinions/findings echo our research, so we still keep the citations.
  • We re-stated the aimed point of this article, and the interaction of lifestyle and pollutants need to be further studied in the future.
  • We have added more infomations, such as WBC counts in table 1.
  • The conclusion paragraph has been rewritten to improve.
  • The advised typo errors have been corrected

Reviewer 3

The manuscript  ijerph-1100727 entitled ‘The Association of White Blood Cells and Air Pollutants– A Population-Based Study’ investigates the relationship between multiple air pollutants and white blood cells counts by means of a population based observational study.

This work analyzes a sample with a considerable number of participants . The selection of the sample as well as the statistical data analysis seem appropriate. However, the presentation and discussion of experimental results could be improved by including the clarification of several aspects. For these reasons, it is recommended the revision of the present version before publication.   

Specific comments:

- Abstract: the wording of several sentences should be revised since it is misleading. For example, the meaning of the first sentence is confused. In Line 23, it is more appropriated the expression: ‘the goal of the study’ instead of ‘the goal is to exam’. The term ‘administrating’ is not correctly employed when referring to a population-based study. Results should be further emphasized including statistical data related to multiple regression analysis.

- Introduction: English language requires substantial changes so that the intended meaning is clear. The effect of particular matter on the count of white blood cells is described through a number of works in human samples, however the presentation of the information could be improved showing more consistent data and not a mere enumeration of studies. References could be further updated. Subscripts in chemical formulas are missing.

- Methods: the procedure for obtaining blood samples should be further described, including the period of time and fasting conditions. An overview of the existing legislation on air quality along with the thresholds and limits for air pollutants should be included in this section or the introduction in order to illustrate the environmental conditions of the studied areas.

- Results and discussion: The presentation and discussion of results could be improved taking into account the following considerations:

  • Table 1 shows sociodemographic data and white cell blood counts classified for each district. Nevertheless, the association between air pollutants and white blood cell counts is neither mentioned nor showed with regard to each specific area in the rest of the tables. Only Table 6 describes changes of white blood cell counts referred explicitly to the whole city. It is not clear the criteria followed for considering the data totally or partially.
  • The expression: ‘short term exposure’ could be confusing, since it implies that blood samples were taken before and after the exposure, or during the period of 7 days considered for the study. However, as mentioned in methods, blood samples were only taken the ‘examination day’ whereas the air pollution data were selected 7 days before that day.
  • Discussion of results according to the simple and multiple regression analysis is not adequately presented. The effect of air pollutants on white blood cells counts is not described and the studies presented are not related with the results obtained in the work.
  • The inclusion of tables 2 and 3 as supplementary material could be considered.
  • Subscripts in some chemical formulas are missing.

Author Response

The manuscript ijerph-1100727 entitled ‘The Association of White Blood Cells and Air Pollutants– A Population-Based Study’ investigates the relationship between multiple air pollutants and white blood cells counts by means of a population based observational study.

This work analyzes a sample with a considerable number of participants. The selection of the sample as well as the statistical data analysis seem appropriate. However, the presentation and discussion of experimental results could be improved by including the clarification of several aspects. For these reasons, it is recommended the revision of the present version before publication.  

Response: Thanks for the comment. Yes, we have revised many parts of the manuscript according to the reviewer’s direction.

Specific comments:

- Abstract: the wording of several sentences should be revised since it is misleading. For example, the meaning of the first sentence is confused. In Line 23, it is more appropriated the expression: ‘the goal of the study’ instead of ‘the goal is to exam’. The term ‘administrating’ is not correctly employed when referring to a population-based study. Results should be further emphasized including statistical data related to multiple regression analysis.

Response: Thanks for the comment. The abstract was revised under the reviewer’s direction.

- Introduction: English language requires substantial changes so that the intended meaning is clear. The effect of particular matter on the count of white blood cells is described through a number of works in human samples, however the presentation of the information could be improved showing more consistent data and not a mere enumeration of studies. References could be further updated. Subscripts in chemical formulas are missing.

Response: Thanks for the comment. The English of introduction section has been revised for improved reading. The references have been updated to 2020. Although some references are old, but they seem to be representative, such as Wan C. Tan et al., 2000, studied the exposure of forest fire; Joel Schwartz, 2001, involved over 20,000 participants in study; Liao, D et al., 2005, found a significant curvilinear association and implies there is a threshold effect; Rückerl et al., 2007, conducted a prospective study; Rich et al., 2012, using a quasi-experiment during the Beijing Olympics, etc. Thus, we still keep them.

The missing of subscripts in chemical formula have been corrected.

- Methods: the procedure for obtaining blood samples should be further described, including the period of time and fasting conditions. An overview of the existing legislation on air quality along with the thresholds and limits for air pollutants should be included in this section or the introduction in order to illustrate the environmental conditions of the studied areas.

Response: Thanks for the comment. The procedure of obtaining blood sample and overview of legislation have been added to the second paragraph of method section, page 3.

- Results and discussion: The presentation and discussion of results could be improved taking into account the following considerations:

Table 1 shows sociodemographic data and white cell blood counts classified for each district. Nevertheless, the association between air pollutants and white blood cell counts is neither mentioned nor showed with regard to each specific area in the rest of the tables. Only Table 6 describes changes of white blood cell counts referred explicitly to the whole city. It is not clear the criteria followed for considering the data totally or partially.

Response: Thanks for the comment. The pollutants regard to each specific district were added in table 1. However, the datasets of air pollutants and WBC in each district were merged for model analysis in the regression step. Readers can check the associations between air pollutant and WBC in tables 3-5. The datasets of air pollutants and WBC in each district were merged for model analysis in the regression steps. Readers can refer to tables 3-5 for the associations between air pollutant and WBC.

The expression: ‘short term exposure’ could be confusing, since it implies that blood samples were taken before and after the exposure, or during the period of 7 days considered for the study. However, as mentioned in methods, blood samples were only taken the ‘examination day’ whereas the air pollution data were selected 7 days before that day.

Response: Thanks for the comment. The expression of ‘short term exposure’ has been modified or deleted, or replaced by ‘during the period of 8 days’, ‘short-term ambient concentration’ in some sentences, under reviewer’s direction to avoid confusing.

Discussion of results according to the simple and multiple regression analysis is not adequately presented. The effect of air pollutants on white blood cells counts is not described and the studies presented are not related with the results obtained in the work.

Response: Thanks for the comment. We have tried our best effort to revise the manuscript. Please read the revised version that we have re-written many parts.

The inclusion of tables 2 and 3 as supplementary material could be considered.

Subscripts in some chemical formulas are missing.

Response: Regarding ‘The inclusion of tables 2 and 3 as supplementary material could be considered… ‘, thanks for the opinion. However, we think that table 2 and table 3 are important for the research presentation. If we omitted these 2 tables or move them to supplement material, the readers would not be convenient to read and understand.

Reviewer 4

Comments

In this manuscript, Hung and colleagues studied the association of white blood cells and air pollutants using environmental pollutant data from air quality monitoring stations and the blood counts data generated by hematology automated analyzer. They found that short-term exposure to air pollutants, especially CO and SO2, related to white blood cell count changes. The study is very interesting and the manuscript is well presented in general, while the authors should address the following concerns before the study can be considerate for publication.

  1. Information of “Methods” is not adequate. Please explain in “method” how “variables of gender, age,.. have been adjusted” (lines 161-162 and 171-172). Please also provide the detailed information for the analysis of “multiple-pollutant model”.
  2. Please indicate the “units” of WBC counts of Table 1 and also indicate whether the units for while blood counts in the manuscript are the same or not.
  3. Line 153-156, please double check if the numbers match the numbers in Table 2.
  4. It will be better if the authors can change Table 2 to Figure so that readers will get more information about the pollutants, including the highest, average, lowest, duration et al.

Author response

Review Report (Reviewer 3)

In this manuscript, Hung and colleagues studied the association of white blood cells and air pollutants using environmental pollutant data from air quality monitoring stations and the blood counts data generated by hematology automated analyzer. They found that short-term exposure to air pollutants, especially CO and SO2, related to white blood cell count changes. The study is very interesting and the manuscript is well presented in general, while the authors should address the following concerns before the study can be considerate for publication.

Information of “Methods” is not adequate. Please explain in “method” how “variables of gender, age,... have been adjusted” (lines 161-162 and 171-172). Please also provide the detailed information for the analysis of “multiple-pollutant model”.

Response: Thanks for the comment. We have added the point in the section of method, page 4. In brief, the multivariate regression equations are listed as below:

  • [single pollutant model]
    Y= β0 + β1(age) + β2(gender) + β3(education) + β4(cigarette smoking) + β5(alcohol consumption) + β6(betel quid use) + β7(individual pollutant);
    Y: WBC counts; the individual pollutant: SO2, NO2, CO, O3, PM10
  • [multiple pollutants model]
    Y= β0 + β1(age) + β2(gender) + β3(education) + β4(cigarette smoking) + β5(alcohol consumption) + β6(betel quid use) + β7(SO2) + β8(NO2) + β9(CO) + β10(O3) + β11(PM10);
    Y: WBC counts

Please indicate the “units” of WBC counts of Table 1 and also indicate whether the units for while blood counts in the manuscript are the same or not.

Response: Thanks for the comment. The “units” of WBC, count/μL, has been added.

Line 153-156, please double check if the numbers match the numbers in Table 2.

Response: Thanks for the comment. Line 153-156 and table 2 were corrected, and add Maximum and minimum.

It will be better if the authors can change Table 2 to Figure so that readers will get more information about the pollutants, including the highest, average, lowest, duration et al.

Response: Thanks for the comment, however, because the units of pollutants are different and the number scale is wide, we think table style can show the picture clearly and easily.

Round 4

Reviewer 2

The Association of White Blood Cells and Air Pollutants– A Population-Based Study
Authors: Shih-Chiang Hung, Hsiao-Yuan Cheng, Chen-Cheng Yang, Chia-I Lin, Chi-Kung Ho, Wen-Huei Lee, Fu-Jen Cheng, Chao-Jui Li, Hung-Yi Chuang

There is no information on how the results were obtained or how the calculations were performed. Please provide formulas.

The Conclusions presented in this article are of no scientific benefits. Argue the conclusions with numerical values.

Author Response

The Association of White Blood Cells and Air Pollutants– A Population-Based Study
Authors: Shih-Chiang Hung, Hsiao-Yuan Cheng, Chen-Cheng Yang, Chia-I Lin, Chi-Kung Ho, Wen-Huei Lee, Fu-Jen Cheng, Chao-Jui Li, Hung-Yi Chuang

There is no information on how the results were obtained or how the calculations were performed. Please provide formulas.

Response: Thanks for the suggestion.

  • The formulas have been added to the section of method. On page 4:
  • The added formulas are:
    [single pollutant model]

Y= β0 + β1(age) + β2(gender) + β3(BMI) + β4(cigarette smoking) + β5(alcohol consumption) + β6(betel quid use) + β7(individual pollutant);

Y: WBC counts; the individual pollutant: SO2, NO2, CO, O3, PM10

  • [multiple pollutants model]

Y= β0 + β1(age) + β2(gender) + β3(BMI) + β4(cigarette smoking) + β5(alcohol consumption) + β6(betel quid use) + β7(SO2) + β8(NO2) + β9(CO) + β10(O3) + β11(PM10);

Y: WBC counts

The Conclusions presented in this article are of no scientific benefits. Argue the conclusions with numerical values.

Response:

  • Thank for this suggestion, the conclusion section has been revised according to reviewer’s advice. On page 12:

In this population-based study, we find air pollutants is associated with WBC counts, particularly SO2, NO2 and CO. Every unit increase of SO2, NO2 and CO, is associated with WBC count change of -33.7, -22.3, and 1431.1, respectively. These would imply that air pollution has impacts on systematic inflammation response. Our research finding is in line with previous studies, supports the postulated link of air pollution to inflammation, which is a common pathway sparking off human diseases. However, this study is local area scale, further prospective, larger-scale (ex. nationwide) studies are recommended. Variables associated with WBC count, such as exercise, lipid pro-files, or another inflammation-related biomarkers could be involved in the future re-search.

Reviewer 3

The quality of the manuscript ijerph-1100727 entitled ‘The Association of White Blood Cells and Air Pollutants– A Population-Based Study’ has been improved. However, several recommendations have not been conveniently addressed:

  • Introduction: subscripts have been correctly revised. Nevertheless, the presentation of the information remains the same and is reported like a list of studies. Although the response of the authors is correct, references has not been changed.
  • Results: regarding the information presented in the Tables, the authors should mention that the the analysis has been carried out for the entire area and has not considered the results obtained for each district separately. A reference to this issue should be included in the text.
  • Discussion: this section does not present significant variations. The combined effect of air pollutants on white blood cell counts as obtained from multiple regression analysis has not been described.
  • English language still requires changes.

Author response

The quality of the manuscript ijerph-1100727 entitled ‘The Association of White Blood Cells and Air Pollutants– A Population-Based Study’ has been improved. However, several recommendations have not been conveniently addressed:

  • Introduction: subscripts have been correctly revised. Nevertheless, the presentation of the information remains the same and is reported like a list of studies. Although the response of the authors is correct, references has not been changed.

Response:

  • Thanks for the suggestion. We have updated more references of recent 2 years, and increasing from 38 to 54 references totally. Of the total references, there were 10 published in 2020, and 7 published in 2019. If there was any important reference we did not cite, we wish the reviewers and editor could help to mention these references please.
  • Results: regarding the information presented in the Tables, the authors should mention that the the analysis has been carried out for the entire area and has not considered the results obtained for each district separately. A reference to this issue should be included in the text.

Response:

  • Thanks for the suggestion. The description of entire areas analysis has been mentioned in this revision page 6. New references were also added. We have updated more references of recent 2 years, and increasing from 38 to 54 references totally. Of the total references, there were 10 published in 2020, and 7 published in 2019.
  • Discussion: this section does not present significant variations. The combined effect of air pollutants on white blood cell counts as obtained from multiple regression analysis has not been described.

Response:

  • Thanks for the suggestion. We have added more description about effects of air pollutants in discussion section, paragraphs 1, 3, 4, and 5 (pages 10-11). And potential variables adjusted in multiple regression analysis have been descripted in discussion section, paragraph 6 (page 12).
  • English language still requires changes.

Response:

  • Thanks for the suggestion. English has been improved for reading.